# Occupational exposure to *Brucella* spp.: A systematic review and meta-analysis

**Carine Rodrigues Pereira**[1], **João Vitor Fernandes Cotrim de Almeida**[1], **Izabela Regina Cardoso de Oliveira**[2], **Luciana Faria de Oliveira**[3], **Luciano José Pereira**[4], **Márcio Gilberto Zangerônimo**[1], **Andrey Pereira Lage**[5], **Elaine Maria Seles Dorneles**[1]*

**1** Departamento de Medicina Veterinária, Universidade Federal de Lavras, Lavras, Minas Gerais, Brazil, **2** Departamento de Estatística, Universidade Federal de Lavras, Lavras, Minas Gerais, Brazil, **3** Programa Nacional de Controle e Erradicação da Brucelose e Tuberculose Animal, Instituto Mineiro de Agropecuária, Belo Horizonte, Minas Gerais, Brazil, **4** Departamento de Ciências da Saúde, Universidade Federal de Lavras, Lavras, Minas Gerais, Brazil, **5** Departamento de Medicina Veterinária Preventiva, Escola de Veterinária, Universidade Federal de Minas Gerais, Belo Horizonte, Minas Gerais, Brazil

* elaine.dorneles@ufla.br

**Data Availability Statement:** All relevant data are within the manuscript and its Supporting Information files.

## Abstract

Brucellosis is a neglected zoonotic disease of remarkable importance worldwide. The focus of this systematic review was to investigate occupational brucellosis and to identify the main infection risks for each group exposed to the pathogen. Seven databases were used to identify papers related to occupational brucellosis: CABI, Cochrane, Pubmed, Scielo, Science Direct, Scopus and Web of Science. The search resulted in 6123 studies, of which 63 were selected using the quality assessment tools guided from National Institutes of Health (NIH) and Case Report Guidelines (CARE). Five different job-related groups were considered greatly exposed to the disease: rural workers, abattoir workers, veterinarians and veterinary assistants, laboratory workers and hunters. The main risk factors and exposure sources involved in the occupational infection observed from the analysis of the articles were direct contact with animal fluids, failure to comply with the use of personal protective equipment, accidental exposure to live attenuated anti-brucellosis vaccines and non-compliance with biosafety standards. *Brucella* species frequently isolated from job-related infection were *Brucella melitensis*, *Brucella abortus*, *Brucella suis* and *Brucella canis*. In addition, a meta-analysis was performed using the case-control studies and demonstrated that animal breeders, laboratory workers and abattoir workers have 3.47 [95% confidence interval (CI); 1.47–8.19] times more chance to become infected with *Brucella* spp. than others individuals that have no contact with the possible sources of infection. This systematic review improved the understanding of the epidemiology of brucellosis as an occupational disease. Rural workers, abattoir workers, veterinarians, laboratory workers and hunters were the groups more exposed to occupational *Brucella* spp. infection. Moreover, it was observed that the lack of knowledge about brucellosis among frequently exposed professionals, in addition to some behaviors, such as negligence in the use of individual and collective protective measures, increases the probability of infection.

**Funding:** This study was supported by Conselho Nacional de Desenvolvimento Científico e Tecnológico (CNPq) (http://cnpq.br/), Coordenação de Aperfeiçoamento de Pessoal de Nível Superior (Capes) (https://www.capes.gov.br/) and Fundação de Amparo à Pesquisa do Estado de Minas Gerais (Fapemig) (https://fapemig.br/pt/). APL, LJP and MGZ are greatful to CNPq for their fellowships. CRP thanks Fapemig for her fellowship. The funders had no role in study design, data collection and analysis, decision to publish, or preparation of the manuscript.

**Competing interests:** The authors have declared that no competing interests exist.

## Author summary

Brucellosis is a zoonotic bacterial infection of major importance worldwide, affecting not only domestic animals but different wildlife species. Due to its ways of transmission, direct or indirect contact with infected animals or their contaminated biological products, the disease exhibits a strong occupational character. This systematic review addressed the main occupations affected by *Brucella* spp. infection, due to the regular exposure to aerosol and contact of non-intact skin (e.g. wounds and abrasion) with infected materials, such as carcasses, viscera and live attenuated anti-brucellosis vaccines. The main risk factors for the disease were identified, as well as the most common forms of exposure to the pathogen. In addition, the most frequently *Brucella* species isolated from farmers, abattoir workers, veterinarians and veterinary technicians, laboratory workers and hunters were also described. The constant contact with the pathogen, the lack of information and instructions to occupational groups exposed, as well as the low adhesion to personal protective equipment in the work environment are determining factors for the occurrence of brucellosis among these individuals.

## Introduction

Brucellosis is one of the most common anthropozoonosis in the world, with approximately 500,000 new human cases reported annually to the World Health Organization (WHO) [1]. Accidental exposure of humans through the ingestion of dairy products made of raw milk, unprotected contact with infected animals or contaminated biological materials, and accidental exposure to anti-*Brucella* spp. vaccines used in veterinary practice are the major forms of disease transmission, which has a strong occupational feature [2,3]. The worker groups most exposed to the pathogen are breeders and animal handlers, butchers, laboratory workers, veterinarians and veterinary assistants, and hunters [4].

In humans, disease caused by infection by bacteria of the genus *Brucella* is characterized by non-specific acute symptoms, such as fever, malaise, chills, weight loss and arthralgia. In some cases, brucellosis can evolve to chronic signs, which can affect a large number of systems and cause osteomyelitis, orchitis and endocarditis, among other manifestations [1,5]. Treatment of the disease is usually long and with strong side effects, intended to control the acute form of the ilness and to prevent the chronic one, with development of sequelae that may incapacitate the individual for work [6]. The administration of two synergistic antibiotics, doxycycline and rifampicin or doxycycline and an aminoglycoside, is normally recommended (among other possible therapies) and the treatment should last a period of at least six weeks [7,8]. Moreover, the discontinuity of chemotherapy is responsible for debilitating complications and relapses. On a global basis, brucellosis is one of the 20 highest-ranked conditions with impact on impoverished people [9]. Damage caused by the disease in individuals' quality of life is intangible and the economic losses attributed to the infection in humans are associated to the costs of hospital treatment, drugs and absence from work due to disabling feature of the disease in its severe form [6]. These damages are more intense in groups frequently exposed to microorganisms of the genus *Brucella*: the Disability-Adjusted Life Year (DALY), a metric that quantifies the burden of mortality and morbidity caused by a disease, were found to be 0.13 [95% uncertainty interval (UI) 0.06–0.18] per thousand persons per year in non-occupational adult and 0.29 [95% UI; 0.08–0.70] per thousand persons per year in occupational population (farmers, abattoir workers and veterinarians) for human brucellosis in India [10], in which one DALY can be thought of as one lost year of "healthy" life.

The prevention of brucellosis transmission among occupations that directly deal with animals or their products relies on effective defensive measures, as the adoption of personal protective equipment (PPE) during activities involving the risk of *Brucella* spp. infection [11]. Manipulation of potentially infected animals, contaminated biological materials and live attenuated anti-brucellosis vaccines are risk factors of remarkable importance for human brucellosis; however, the more detailed knowledge about particular risk factors to each occupation, as well as the measurement of these risks is still scarce. In fact, there is a need for more accurate data on the epidemiology of job-related brucellosis to allow the implementation of more effective preventive measures, which will reduce the impact of the disease in groups exposed by their work activities. The availability of these information could also be translated into health protection behaviors among susceptible professionals. Thus, the aims of this systematic review were (i) to identify high quality studies that reported and evaluated occupational exposure to brucellosis, (ii) to evaluate the main risk factors of each exposed group (rural workers, abattoir workers, laboratory workerss, veterinarians, veterinary technicians and hunters), and (iii) to estimate, by means of a meta-analysis, the odds of individuals occupationally exposed to *Brucella* spp. become infected, compared to individuals not exposed to direct animal contact or their biological fluids.

## Methods

The guidelines of PRISMA statement (Preferred Reported Items for Systematic Reviews and Meta-Analyses) [12] were formally adopted in this review and can be seen in additional file 1 (S1 Appendix).

### Search strategy

The search was conducted on May 16, 2018, without any date or country restriction. All the keywords were investigated within title, abstract and full text sections in the following databases: CABI, Cochrane, Pubmed, Scielo, Science Direct, Scopus and Web of Science. The PICO (population, intervention, comparison and outcome) used for the search were: veterinarians, laboratory workers, farmers and abattoir workers (population), exposure to *Brucella* spp. (intervention), occupational and job-related (comparison) and brucellosis (outcome). An overview of the search terms is shown in additional file 2 (S2 Appendix).

### Selection of the studies

The literature search returned original papers published between 1931 and 2018. The database content was exported to Endnote X7.8, checked and cleaned for duplicates [13]. In the second stage, for those studies selected based on their titles (CRP), two reviewers independently evaluated the abstract of each paper (CRP and JVFCA). Subsequently, full text of the papers selected based on the abstracts were evaluated by two reviewers in terms of its relevance and by means of inclusion/exclusion criteria (CRP and JVFCA). When these reviewers disagreed over the inclusion or exclusion of a paper, a third reviewer was responsible for the final decision (EMSD). Further, the reference lists of selected papers were reviewed in order to find pertinent studies not identified during the initial search.

### Inclusion and exclusion criteria

The following characteristics were considered for the inclusion of articles: (i) articles focusing on *Brucella* spp., (ii) concerning occupational exposure to *Brucella* spp. or to brucellosis infection in humans and (iii) written in English, French, Spanish and Portuguese. Articles aiming

on (i) animal brucellosis, (ii) genetics, immunology, microbiology or drug therapy were excluded. Full inclusion and exclusion criteria are shown in additional file 3 (S3 Appendix).

### Type of studies

Original papers, using quantitative or qualitative data, as cohort, case-control, cross-sectional and case series studies and case reports were included. Reviews were excluded and their references were identified through manual search in order to find relevant articles.

### Data extraction and quality assessment

Data were extracted from papers by one of the reviewers (CRP) and were subsequently checked for accuracy by other reviewer (JVFCA). Disagreements regarding data extraction among reviewers were solved by consensus. Extracted data included: first author, geographic location, study period, target population, number of positive individuals, study design, diagnostic method and cutoff values, *Brucella* species isolated, identification of occupational exposure, predictors of transmission, potential risks factors for the development of brucellosis among high-risk groups and possible molecular confirmation from the source of infection. The case definitions described in each study by the respective authors were considered. The quality of cohort, case-control, cross-sectional and case series studies was evaluated using the quality assessment tools from the National Heart, Lung and Blood Institute (NHI) and CARE (Case Report) checklist was used for quality assessment of case reports [14].

### Meta-analysis

Case-control studies were selected to estimate the odds of individuals occupationally exposed to *Brucella* spp. become infected, compared to individuals without occupational risk. The homogeneity among the studies was verified using Cochrane's Q test, and the total variability related to among-study variations was reflected in the $\tau^2$, which was estimated by the DerSimoninan-Laird method [15]. The pooled odds ratio (OR) of the studies was obtained through a random effect modeling and by the adoption of the Mantel-Haenszel estimator [15]. The meta-analysis was performed with R statistical software 3.5.2 [16], using the meta package [17].

## Results

The search strategy adopted identified a total of 6123 papers; 454 duplicates were excluded, and 239 full-texts were assessed for eligibility. Subsequently, 63 papers from 1962 to 2018 were included in quality level assessment and data synthesis appraisal, after a thorough review (Fig 1). The background characteristics (geographic location, study period, target population, number of positive individuals, study design, diagnostic method and cutoff values, *Brucella* species isolated, identification of occupational exposure, predictors of transmission, potential risks factors for the development of brucellosis among high-risk groups and possible molecular confirmation from the source of infection) were identified in these articles and are shown in additional file 4 (S4 Appendix).

The assessment of geographical origin on selected job-related brucellosis papers showed that seven studies were from Africa, seventeen from the Americas, twenty-two from Asia and seventeen from Europe (Fig 2A). Regarding to the year of publication, except for the 1970s, the number of studies published about human brucellosis with occupational feature increased every decade (Fig 2B). Indirect methods, as agglutination tests, indirect-ELISA, 2-mercaptoethanol, complement fixation, among others, were the main tests used to human brucellosis

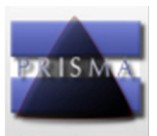

**PRISMA Flow Diagram**

**Identification**

Records identified through database searching (n = 6123)

Additional records identified through manual search in the reference of review articles (n = 40)

**Screening**

Records after duplicates removed (n = 5709)

Records screened (n = 5709)

Records excluded (n = 5422)

**Eligibility**

Full-text articles assessed for eligibility (n = 239)

Full-text articles excluded, with reasons (n = 176):
- Inadequate statistical application (n = 100)
- Poorly described methodology (n = 15)
- Book or review (n = 36)
- Full text not accessed (n = 25)

**Included**

Studies included in qualitative synthesis (n = 63)

Studies included in quantitative synthesis (meta-analysis) (n = 3)

**Fig 1. PRISMA Flow diagram of selected studies.**

diagnosis in the studies, which observed an overall of 1432 individuals occupationally infected. Moreover, the use of direct methods for the diagnosis, such as isolation and polymerase chain reaction (PCR), also revealed 112 positive individuals being infected with *Brucella melitensis*, *Brucella suis*, *Brucella abortus* and *Brucella canis*, shown in additional file 5 (S5 Appendix). The Fig 3 shows the distribution of brucellosis cases by country according to occupational group affected (a) and the *Brucella* species most frequently identified (b).

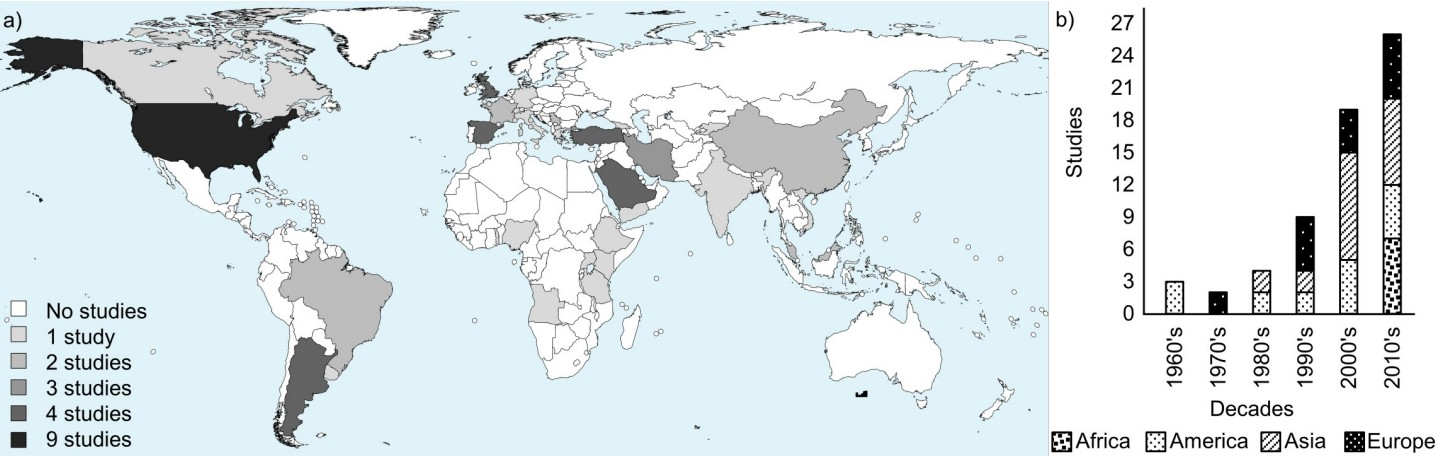

**Fig 2. Geographical and temporal distribution of the selected articles included in the present study.** (a) Distribution and frequency of occupational brucellosis studies published by country (performed with aid of online dataset: https://commons.wikimedia.org/wiki/Atlas_of_the_world). (b) Distribution and frequency of occupational brucellosis studies published by continent and decade, from 1962 to 2018.

## Rural workers

Farmers, shepherds and livestock breeders were the leading groups affected by brucellosis, with 870 positive individuals described in twenty-four studies [2,18–40], of which the most part was carried out in Asia (n = 549), Europe (n = 180), Africa (n = 107) and the minority in America (n = 34). Direct contact with potentially infected cattle, goats and sheep during labor activities, such as calving, barn cleaning and herd vaccination, were described in the studies as potential sources of infection of *Brucella* spp. to rural workers (Table 1). Irrefutable evidence of animal-to-human brucellosis transmission was observed by a study conducted in Argentina, in which the same genotype of *B. melitensis* was observed in milk (n = 17) and colostrum (n = 11) samples from goats and in rural workers (n = 14) who lived near the animals [38]. Moreover, another study also identified that Livestock aborted remains from production animals were abandoned in the pasture and eventually ingested by dogs and pigs, in some properties in Angola [33].

## Abattoir workers

A total of 292 individuals working in slaughterhouses were described as brucellosis-positive in fourteen articles [19,20,23,33,34,41–49]. Most of those individuals were from America (n = 162), Africa (n = 60), and Europe (n = 37) and the minority from Asia (n = 33). The main type of pathogen exposure reported was contact with animal fluids, aborted fetus, placenta and viscera. Accidental contact with those materials was described in three studies: in Spain and Ethiopia, 12.26% (13/106) and 48.72% (76/156) of slaughterhouse workers, respectively, reported cutting themselves with dirty sharp blades [45,46], and in China, 100.00% (3/3) of pharmaceutical employees, who worked processing sheep placenta, reported having splashed animal fluids on their faces [47]. The great occurrence of direct contact with biological contaminated fluids aroused the interest of several authors to understand which PPE were used or not by this group of professionals (Table 2).

## Veterinarians and veterinary assistants

Veterinarians and veterinary assistants showed to be largely exposed to *Brucella* spp., totalizing 189 individuals with positive diagnostic of brucellosis. Those infections, probably related to

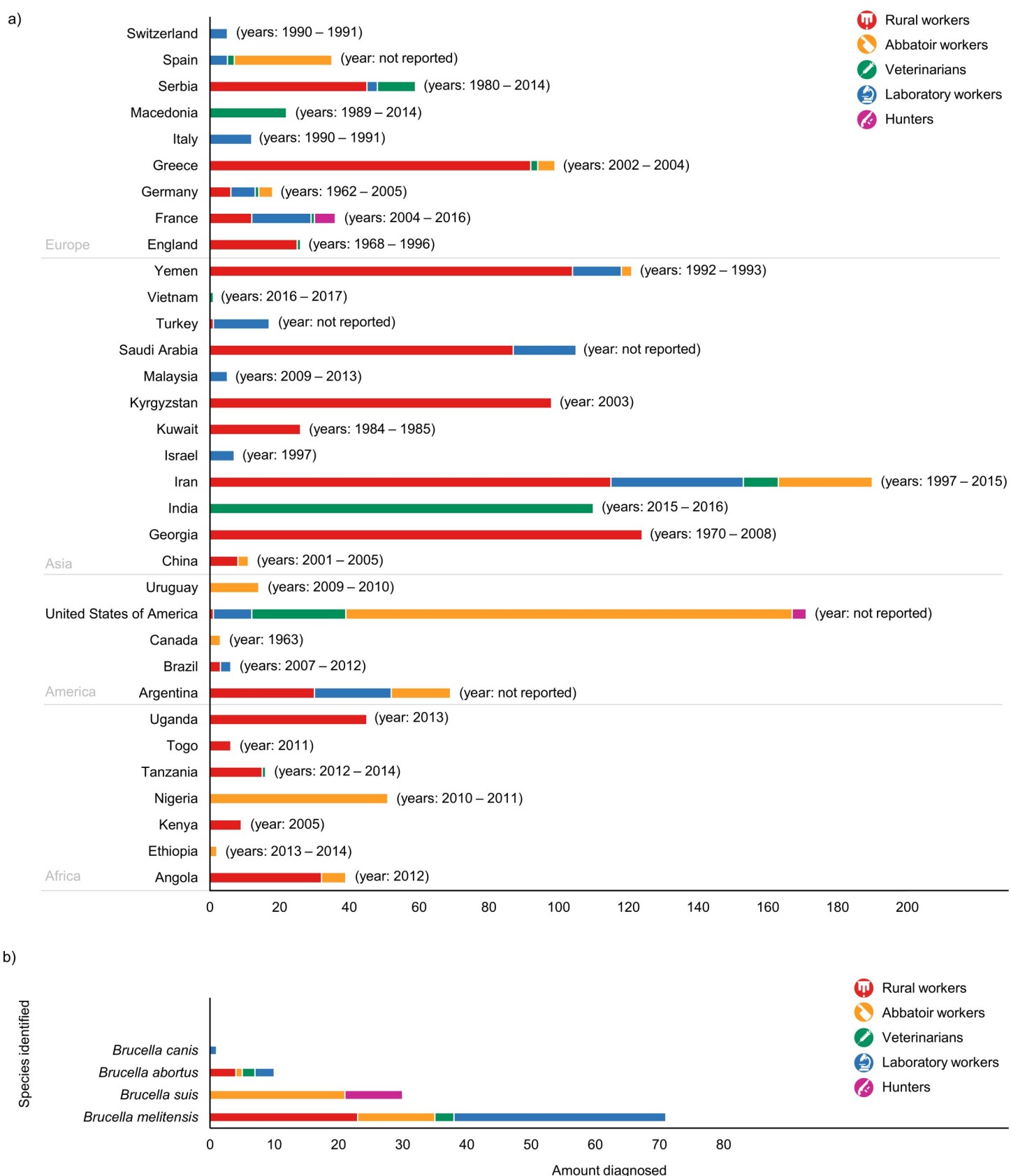

**Fig 3.** Distribution of occupations affected by occupational brucellosis by country, including the and time period when the studies were performed, selected by this systematic review (a) and the *Brucella* species identified through direct diagnostic methods (b).

their occupational activities, were reported by fifteen articles [2,19,23,24,28,31,35,43,50–56], mostly from Asia (n = 121), Europe (n = 40), and Americas (n = 27) and the minority from Africa (n = 1). Manipulation of live attenuated anti-brucellosis vaccines, described in seven studies, was the most reported exposure source (Table 3). Of these, three were able to establish an epidemiological link between the vaccine strain and the strain responsible for the infection in the veterinarians: *B. abortus* strain RB51 was isolated from a surgical wound three days after a self-inoculation [2]; *B. abortus* strain 19 was cultured from a discharge, from the injection site, obtained on the eighth day after a needlestick injury [54]; and *B. melitensis* strain REV-1 was isolated from blood cultures of two veterinarians, several months after the accidental exposure [51]. In addition to this type of exposure, veterinarians and veterinary assistants also reported to perform other activities associated with a high risk of infection, such as attending parturitions and infertility cases, and handling aborted fetus, retained placenta and stillbirths [23,24,31,50]. Furthermore, the use of PPE in some cases was considered inadequate [43,56].

## Laboratory workers

Brucellosis related to laboratory practices was largely reported: 24 papers described this transmission in 183 individuals, of which the majority was from Asia (n = 98) and Europe (n = 49) and the minority from the Americas (n = 36) [19,20,28,31,57–76]. The main factors possibly related to the infection were working outside a safety cabinet, being at the laboratory during or after an accident, failure suspecting brucellosis as a possible diagnosis and sniffing culture plates (Table 4). Two papers reported infection of individuals working outside a laboratory facility, but in indirectly related departments with the presence of *Brucella* spp. positive cultures within the environment. The first case was in a S19 manufacturing plant, where 21 workers were infected probably by the vaccine strain, in Argentina [74]; whereas, the second occurred in a waste treatment plant, where an employee stuck his foot in a needle contaminated with the *B. suis* biovar 1 reference strain 1330 [59] identified by molecular genotyping methods. The epidemiological link (biotyping) between the source of accidental exposure and the patient's isolate was also established in other reports of brucellosis among laboratory

**Table 1. Farm animal species related to occupational brucellosis transmission among infected rural workers.**

| Study | Year† | Country | Total of workers | Contact | |
|---|---|---|---|---|---|
| | | | | Cattle | Small ruminants |
| [2] | 1998–1999 | USA | 1 | 1 | 0 |
| [22] | 2013 | Uganda | 19 | 0 | 19 |
| [26] | 2007 | Brazil | 2 | 2 | 0 |
| [31] | 2004–2013 | France | 11 | 11 | NR |
| [33] | 2012 | Angola | 32 | 32 | NR |
| [36] | 1969 | England | 1 | 1 | 0 |
| [38] | Not reported | Argentina | 32 | 0 | 32 |
| [39] | 1968–1969 | England | 1 | 1 | 0 |
| | | Total | 99 (100.00%) | 48 (48.48%) | 51 (51.52%) |

NR = not reported; USA = United States of America

† = Year of sampling

**Table 2. Not use of personal protective equipment (PPE) among slaughterhouse workers occupationally infected by *Brucella* spp.**

| Study | Year† | Country | Total of workers | PPE not used | | | | |
|---|---|---|---|---|---|---|---|---|
| | | | | Gloves | Masks | Goggles | Boots | Apron |
| [41] | 2010–2011 | Nigeria | 54 | 2 | NR | NR | NR | NR |
| [43] | 2014–2015 | Iran | 198 | 25 | 82 | 20 | 113 | 101 |
| [44] | 2009–2010 | Uruguay | 14 | NR | NR | 0 | NR | NR |
| [45] | 1998–1999 | Spain | 28 | 19 | 18 | 16 | NR | NR |
| [46] | 2013–2014 | Ethiopia | 156 | 29 | NR | NR | NR | NR |
| [47] | 2005 | China | 3 | 3 | 3 | NR | NR | NR |
| [48] | 2014–2015 | Argentina | 17 | 0 | 0 | 0 | NR | NR |
| | | Total* | 470 | 78/456 (17.11%) | 103/246 (41.87%) | 36/257 (14.01%) | 113/198 (57.07%) | 101/198 (51.01%) |

NR = not reported

*the percentage was calculated based on the total individuals interviewed about PPE

† = Year of sampling

technicians in Switzerland (*B. melitensis* biovar 3) and Italy (*B. abortus* biovar 1) [62,63]. Moreover, the same biovar was also identified in 8 laboratory workers, during an brucellosis outbreak in the United States of America [73]. Additionally, in France, the occupational brucellosis from 2004 to 2013 represented 46% of domestic cases (all laboratory exposure), and for 94.1% of the brucellosis-positive patients the respective paired strain was identified at molecular level [31].

## Hunters

Job-related exposure was described in hunters in three papers, totalizing 10 infected individuals, from America and Europe [77–79]. Contact with animal fluid was reported, and in France, *B. suis* biovar 2 was isolated from six hunters, all of whom reported not using any type of personal protective equipment while eviscerating the carcasses of slaughtered animals [79]. Furthermore, a frozen sausage and a tenderloin, from a feral swine hunted by two men (USA), were positive for *B. suis* isolation, and had multiple-locus variable-number of tandem repeats analysis (MLVA) signatures identical to a *B. suis* strain isolated from one of the patients [77].

**Table 3. Adverse events or occupational brucellosis in veterinarians and veterinary assistants associated with accidental exposure to anti- *Brucella* spp. live attenuated vaccines.**

| Study | Year† | Country | Total of workers | Vaccine strain | | |
|---|---|---|---|---|---|---|
| | | | | RB51 | S19 | REV-1 |
| [2] | 1998–1999 | USA | 19 | 19 | 0 | 0 |
| [18] | 1970–1973 1988–1889 2004–2008 | Georgia | 1 | NR | NR | NR |
| [23] | 2002–2004 | Greece | 41 | 0 | 0 | 41 |
| [51] | Not reported | Spain | 2 | 0 | 0 | 2 |
| [54] | Not reported | USA | 1 | 0 | 1 | 0 |
| [55] | 1984 | USA | 1 | 0 | 1 | 0 |
| [56] | 2015–2016 | India | 5 | 0 | 5 | 0 |
| | | Total | 70 | 19 (27.14%) | 7 (10.00%) | 43 (61.43%) |

NR = not reported; USA = United States of America

† = Year of sampling

**Table 4. Types of exposure associated with occupational transmission of *Brucella* spp. reported by infected laboratory workers.**

| Study | Year† | Country | Total of workers | Possible cause of infection | | | |
|---|---|---|---|---|---|---|---|
| | | | | Work Outside safety cabinet | Accident reported | Wrong diagnostic* | Sniffed plates |
| [57] | Not reported | Saudi Arabia | 4 | 2 | 2 | 0 | 0 |
| [59] | 2014 | Spain | 1 | 0 | 1 | 0 | 0 |
| [60] | Not reported | Turkey | 3 | 0 | 0 | 0 | 3 |
| [62] | 1990–1991 | Italy | 12 | 0 | 12 | 0 | 0 |
| [63] | 1990–1991 | Switzerland | 2 | 0 | 0 | 2 | 0 |
| [65] | 1983–1990 | Saudi Arabia | 2 | 1 | 0 | 0 | 1 |
| [66] | 1998 | Spain | 4 | 4 | 0 | 0 | 0 |
| [67] | 1991–2000 | Saudi Arabia | 1 | 0 | 0 | 1 | 1 |
| [68] | 2001–2002 | USA | 2 | 2 | 0 | 0 | 0 |
| [70] | 2012 | Brazil | 3 | 0 | 3 | 0 | 0 |
| [72] | 1979 | USA | 1 | 1 | 0 | 0 | 0 |
| [74] | 1999–2006 | Argentina | 5 | 0 | 5 | 0 | 0 |
| [75] | Not reported | Argentina | 1 | 1 | 0 | 0 | 0 |
| | | Total | 42 | 11 (26.19%) | 23 (54.76%) | 3 (7.14%) | 5 (11.90%) |

USA = United States of America

* = Brucellosis not included as possible diagnosis by the clinician

† = Year of sampling

## Meta-analysis

Individuals who perform risky labor activities, such as farming, or employees from slaughter-houses and laboratories showed 3.47 [95% confidence interval (CI); 1.47–8.19] times more chance to become infected with *Brucella* strains than people who develop other occupational activities (Fig 4).

## Discussion

Brucellosis is a worldwide widespread disease of great importance to public health and has a strong occupational character, with certain professions being more commonly affected by the disease [4]. Therefore, the efforts of this systematic review and meta-analysis were focused on the understanding of the main risk factors associated with occupational brucellosis among occupations considered to be more exposed to the agents. Our findings showed a greater chance of infection among field occupations that have direct contact with animals and their products, as well as indicated the main situations of risk and behaviors associated with

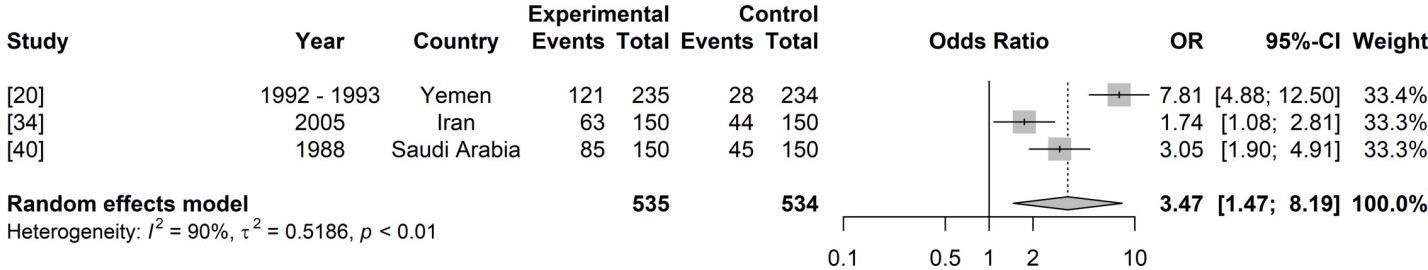

**Fig 4. Forest plot of odds ratio for brucellosis among risk work groups (animal breeders, farmers, abattoir workers and laboratory workers) exposed and other individuals not occupationally exposed to *Brucella* spp. during their labor activities.** Year indicates the period in which study was performed.

infection for each evaluated profession. Information provided by this study is essential to design strategies to minimize the occurrence of occupational brucellosis and to guide specific health protection behaviors to people occupationally exposed.

Although brucellosis is a widespread zoonotic disease, no high-quality studies concerning occupational cases from Oceania were selected, which could be explained by the low occurrence of the disease in the region [1]. Likewise, the differences in the number and emergence of publications among the continents may be due to divergences in the structure of animal and human brucellosis surveillance systems and in the epidemiological situation of the diseases in animals (Fig 2B), since animal brucellosis precede and are closely associated with human brucellosis, especially occupational [80]. Moreover, the increased amount of publications from the 1980's could be associated with the growing importance of the disease in humans and the development of new diagnostic techniques. In fact, the oldest publications selected were from countries that have implemented their animal brucellosis control and prevention programs in the 1910s, 1920s and 1930s, such as the United States and Canada, in the Americas, and Great Britain, in Western Europe [81–83]. On the other hand, some countries in Asia, Latin America and Africa, although presenting endemic animal brucellosis, have not yet reached satisfactory levels of disease control and often report insufficient data on the true prevalence of the infection in humans and animals. Additionally, in those regions, poor interaction between human and veterinary medicine are generally observed [6,84], which could explain the later appearance of scientific publications from those areas among the selected papers. However, it is very important to mention that the number of infected individuals and the number of papers published by country do not have a direct relationship with the actual prevalence of occupational brucellosis in that locality, but is more related to scientific interests of local researchers. In fact, USA showed the biggest number of studies published, although is one of the countries with the lowest incidence of human brucellosis in the world [1].

The indirect methods were mostly common used for the diagnosis of brucellosis, which could be attributed to the lower cost of serologic tests compared to PCR and microorganism isolation, as well as to the safety issues and time saving process compared to bacterial culture [85–87]. Even though not widely used, direct methods have the great advantage of being able to identify the *Brucella* species responsible for the infection, supporting a better understanding of the etiopathogenesis of the disease among the different occupational groups included in this study.

Rural workers are among the group most affected by brucellosis, mainly caused *B. melitensis*, totaling 27 individuals with direct diagnosis of isolation and identification of the *Brucella* species, among the 870 cases observed in this group (Fig 3A). These results are especially important to public health, since *B. melitensis* is one of the most pathogenic and the most prevalent species of *Brucella* spp. for humans [88], and the disease may progress to the development of debilitating symptoms, with severe involvement of multiple organs and systems, and high cost of hospitalization due to the prolonged therapy recommended [89]. The close contact of rural workers with small ruminants, preferred hosts for *B. melitensis*, was identified as the main form of acquisition of the disease among these individuals (Table 1), which has been confirmed by the identification of a high genetic similarity between *B. melitensis* strains isolated from occupationally infected workers and from goat milk samples [38].

The second group most affected by occupational brucellosis (n = 292), mainly by *B. suis* (n = 21), followed by *B. melitensis* (n = 12) and *B. abortus* (n = 1) (Fig 3A), were butchers and abattoir workers, probably due to the regular manipulation of sharp objects and to close contact with potentially infected animals and their organs. Airborne and conjunctival routes were considered important to the transmission of brucellosis among this group [90], especially in closed places, such as slaughterhouses, in which direct contact with contaminated viscera and

secretions occurs. The hazard was increased when prophylactic measures were not properly adopted, as highlighted by the low adherence of PPE use, such as gloves, mask, googles, boots and apron (Table 2). In addition, the low educational level of abattoir workers, as well as insufficient knowledge about brucellosis, particularly on its transmission and clinical signs, increases the risk of these professionals becoming infected and reinforce the importance of implementing educational measures to advise about the need to use PPE and to increase the knowledge of brucellosis symptoms and transmission [41,46,47].

Subsequently, veterinarians and veterinary assistants comprised the third occupational group most affected by brucellosis. In addition to contact with secretions and excretions of potentially infected animals, activities inherent to their work [56], these individuals are the ones with the most important exposure to *Brucella* spp. live attenuated vaccines (REV-1, S19 and RB51) (Table 3), which are a source of the infection for humans [2]. Accidental exposures to brucellosis live attenuated vaccines are especially important when they occur with RB51, since antibodies against this strain are not detected by routine serological tests and RB51 is resistant to rifampicin, one of the preferential drugs to treat human brucellosis [91]. In fact, the accidental exposure to brucellosis vaccines has great significance to brucellosis cases among veterinarians and assistants (n = 189), being confirmed by direct diagnostic methods that revealed *Brucella* spp. infection caused by *B. melitensis* (n = 3) and *B. abortus* (n = 2) vaccine strains [2,51,53,54]. These findings strengthen the importance of use PPE not only in the care of animals, but also during the vaccination procedures.

Laboratory workers represent the fourth group most affected by the *Brucella* spp. infection due to their labor activities. In fact, the highest incidences of laboratory-acquired infections were associated with *Brucella* species [92]. Interestingly, this group (n = 183) showed the greatest number of species isolated: *B. melitensis* (n = 33), *B. abortus* (n = 3) and *B. canis* (n = 1) (Fig 3A), which could be explained by the wide variety of clinical specimens that are often handled by those professionals in the diagnostic routine. Moreover, it must be considered that this group had the largest number of *Brucella* spp. strains isolated and identified among the occupations evaluated, probably due to greater access to direct methods of diagnosis in the environments where they were occupationally exposed. The isolation of *B. canis* in a worker in this occupational group is noteworthy, as it was caused by the M- strain, a strain used for the serologic diagnosis of canine brucellosis that has reduced virulence in dogs [75]. Nonetheless, albeit generally well instructed about the risk of contracting a zoonotic infection during labor activities, many laboratory workers adopted attitudes that put their own health and of their colleagues at risk, as work outside safety cabinet and sniff the plates (Table 4). *Brucella* spp. cultures must only be handled in laboratories with biosafety level 3 or higher [92]; however, due the lack of specificity of the clinical signs caused by the disease, associated with the effectiveness of public policies in some European countries [19,31], where brucellosis occurs primarily among travelers, many physicians rarely raise the hypothesis of brucellosis when sending biological samples for laboratory analysis, leading to exposure to the agent during manipulation of the clinical material by the microbiologist [93]. Misidentification of the organism also happens and puts the health of individuals who manipulate cultures at risk [63,67]. Furthermore, accidents, as damages in the biological safety cabinet or the centrifuge, may also occur in the biosafety level 3 laboratory, reinforcing that training activities to the staff must be periodically carried out in order to ensure cautious manipulation of positive *Brucella* spp. cultures, as well as regular laboratorial equipment maintenance [93]. Indeed, adherence to rigorous infection control measures are important from the receipt to the proper disposal of biological materials, since in this occupational group not only microbiologists but also people working in the laboratory waste processing were affected [59].

The occupation with the lowest number of infected individuals identified was the group of hunters (n = 10), which differently from the previous groups exhibited exclusively *B*. *suis* isolates (n = 9) (Fig 3A). Hunting, a widespread activity in United States of America and in some European countries, such as France, is often associated with the primary route of transmission for *B*. *suis* in humans: through the contact and dressing of carcasses [5,90]. Therefore, the presence of bacteria in the muscular tissues of boars is sufficient to cause infection in humans, especially when carried out without the proper use of individual protection measures.

The occupational character of human brucellosis is supported by the results generated from the meta-analysis of 3 case-control studies, which showed that animal breeders, laboratory workers and abattoir workers were significant more likely to become infected with *Brucella* spp. strains than people who develop other job-related activities (OR 3.47; 95% CI: 1.47 to 8.19) (Fig 4). The low number of selected studies with a case control design (n = 3) observed among the articles resulted in the small number of high-quality papers eligible for meta-analysis. It occurred because of the impossibility of access to data of exposed and non exposed individuals. However, it is important to take into account that despite the low number of studies used in the meta-analysis, the total number of individuals analyzed (n = 1069) and those with occupational brucellosis (n = 269) was considerable, supporting the robust results observed (Fig 4). Those data revealed the weight of exposure to *Brucella* spp. during labor activities for the occurrence of human brucellosis, which is essential to take into account for the design of strategies to minimize its occurrence.

The greatest strengths of this paper are that it is based on the PRISMA statement (as recommended for conducting systematic reviews), that the search was performed in seven scientifically validated and large databases and that the quality assessment of papers were through NIH and CARE guidelines, which allowed the accomplishment of meta-analysis and mitigated possible bias among studies. On the other hand, there are some limitations, such as the differences among case definitions and diagnostic capacity of different studies, especially due to the diversity of diagnostic techniques employed, (see information on additional file 5 –S5 Appendix). Furthermore, some papers (n = 25) were not available despite all efforts through the university databases, scientific social media and request.

The lack of accurate information on the quantification and peculiarities of the risk of brucellosis in each occupational group makes it difficult to direct public resources for the control and prevention of brucellosis in individuals most likely to present the disease, especially in a context with several other demands which also require a portion of the available funds, already limited. In this context, this systematic review provided a meticulous understanding of the risk factors peculiar to each of the main occupations (farmers, slaughterhouses, veterinarians, laboratories and hunters) closely related to *Brucella* spp. infection. Our results also revealed the great lack of information from these occupational groups on the importance of applying preventive measures to minimize the risk of transmission of brucellosis during work. In addition, through meta-analysis it was possible not only to confirm the occupational character of brucellosis, widely recognized, but also to quantify this risk in an unprecedented way in the scientific literature through the calculation of odds ratios, systematically compiling studies so far dispersed in the literature. These data on human cases of occupational brucellosis can be used as a first step towards adopting a One Health approach, which is an interdisciplinary collaboration that aims to reduce the occurrence of zoonotic diseases in humans, through the prevention of such diseases in animals [94]. Thus, the control of brucellosis could be conducted more efficiently and strategically, in order to reduce the incidence of the disease not only in humans, but also in animals and in the environment.

In conclusion, our results reinforced the strong occupational character of human brucellosis, especially among rural workers, slaughterers, veterinarians and veterinary assistants,

laboratory workers and hunters, and revealed the specific risks associated with each occupation. Moreover, it was observed that the lack of knowledge about brucellosis among frequently exposed workers, in addition to some behaviors, such as negligence in the use of individual and collective protective measures, increased the probability of infection.

## Supporting information

**S1 Appendix. PRISMA checklist.**
(DOCX)

**S2 Appendix. Extensive overview of search terms.**
(DOCX)

**S3 Appendix. Inclusion and exclusion criteria for selection of articles.**
(DOCX)

**S4 Appendix. Studies describing occupational human infection by *Brucella* spp..**
(DOCX)

**S5 Appendix. Number of *Brucella* isolation per worker category.**
(DOCX)

## Acknowledgments

The authors are extremely grateful to Nammalwar Sriranganathan for his techinical support.

## Author Contributions

**Conceptualization:** Carine Rodrigues Pereira, Luciana Faria de Oliveira, Andrey Pereira Lage, Elaine Maria Seles Dorneles.

**Data curation:** Carine Rodrigues Pereira, João Vitor Fernandes Cotrim de Almeida, Luciano José Pereira, Márcio Gilberto Zangerônimo.

**Formal analysis:** Carine Rodrigues Pereira, João Vitor Fernandes Cotrim de Almeida, Izabela Regina Cardoso de Oliveira, Luciana Faria de Oliveira, Andrey Pereira Lage, Elaine Maria Seles Dorneles.

**Funding acquisition:** Luciano José Pereira, Márcio Gilberto Zangerônimo.

**Investigation:** Carine Rodrigues Pereira, João Vitor Fernandes Cotrim de Almeida, Izabela Regina Cardoso de Oliveira, Luciana Faria de Oliveira, Elaine Maria Seles Dorneles.

**Methodology:** Carine Rodrigues Pereira, João Vitor Fernandes Cotrim de Almeida, Izabela Regina Cardoso de Oliveira, Elaine Maria Seles Dorneles.

**Project administration:** Elaine Maria Seles Dorneles.

**Software:** Izabela Regina Cardoso de Oliveira, Andrey Pereira Lage.

**Supervision:** Luciana Faria de Oliveira, Andrey Pereira Lage, Elaine Maria Seles Dorneles.

**Validation:** Izabela Regina Cardoso de Oliveira, Andrey Pereira Lage, Elaine Maria Seles Dorneles.

**Visualization:** Carine Rodrigues Pereira, Elaine Maria Seles Dorneles.

**Writing – original draft:** Carine Rodrigues Pereira.

**Writing – review & editing:** João Vitor Fernandes Cotrim de Almeida, Luciana Faria de Oliveira, Luciano José Pereira, Márcio Gilberto Zangerônimo, Andrey Pereira Lage, Elaine Maria Seles Dorneles.

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
