## [Decision Letter · Decision Letter 0]

14 Jan 2020

Dear Dr. Dorneles:

Thank you very much for submitting your manuscript "Occupational exposure to human brucellosis infection: a systematic review and meta-analysis" (#PNTD-D-19-01642) for review by PLOS Neglected Tropical Diseases. Your manuscript was fully evaluated at the editorial level and by independent peer reviewers. The reviewers appreciated the attention to an important problem, but raised some substantial concerns about the manuscript as it currently stands. These issues must be addressed before we would be willing to consider a revised version of your study. We cannot, of course, promise publication at that time.

We therefore ask you to modify the manuscript according to the review recommendations before we can consider your manuscript for acceptance. Your revisions should address the specific points made by each reviewer. 

When you are ready to resubmit, please be prepared to upload the following:

(1) A letter containing a detailed list of your responses to the review comments and a description of the changes you have made in the manuscript.

(2) Two versions of the manuscript: one with either highlights or tracked changes denoting where the text has been changed (uploaded as a "Revised Article with Changes Highlighted" file); the other a clean version (uploaded as the article file).

(3) If available, a striking still image (a new image if one is available or an existing one from within your manuscript). If your manuscript is accepted for publication, this image may be featured on our website. Images should ideally be high resolution, eye-catching, single panel images; where one is available, please use 'add file' at the time of resubmission and select 'striking image' as the file type. 

Please provide a short caption, including credits, uploaded as a separate "Other" file. If your image is from someone other than yourself, please ensure that the artist has read and agreed to the terms and conditions of the Creative Commons Attribution License at http://journals.plos.org/plosntds/s/content-license (NOTE: we cannot publish copyrighted images). 

(4) If applicable, we encourage you to add a list of accession numbers/ID numbers for genes and proteins mentioned in the text (these should be listed as a paragraph at the end of the manuscript). You can supply accession numbers for any database, so long as the database is publicly accessible and stable. Examples include LocusLink and SwissProt.

(5) To enhance the reproducibility of your results, we recommend that you deposit your laboratory protocols in protocols.io, where a protocol can be assigned its own identifier (DOI) such that it can be cited independently in the future. For instructions see http://journals.plos.org/plosntds/s/submission-guidelines#loc-methods

While revising your submission, please upload your figure files to the Preflight Analysis and Conversion Engine (PACE) digital diagnostic tool, https://pacev2.apexcovantage.com/ PACE helps ensure that figures meet PLOS requirements. To use PACE, you must first register as a user. Then, login and navigate to the UPLOAD tab, where you will find detailed instructions on how to use the tool. If you encounter any issues or have any questions when using PACE, please email us at figures@plos.org.

We hope to receive your revised manuscript by Mar 14 2020 11:59PM. If you anticipate any delay in its return, we ask that you let us know the expected resubmission date by replying to this email.

To submit a revision, go to https://www.editorialmanager.com/pntd/ and log in as an Author. You will see a menu item call Submission Needing Revision. You will find your submission record there. 

Sincerely,

Tao Lin, DVM, MSc

Associate Editor

Elsio Wunder Jr

Deputy Editor

Reviewer's Responses to Questions

**Key Review Criteria Required for Acceptance?**

**Methods**

-Are the objectives of the study clearly articulated with a clear testable hypothesis stated?

-Is the study design appropriate to address the stated objectives?

-Is the population clearly described and appropriate for the hypothesis being tested?

-Is the sample size sufficient to ensure adequate power to address the hypothesis being tested?

-Were correct statistical analysis used to support conclusions?

-Are there concerns about ethical or regulatory requirements being met?

Reviewer #1: PRISMA recommendations have been proficiently followed

Reviewer #2: (No Response)

**Results**

-Does the analysis presented match the analysis plan?

-Are the results clearly and completely presented?

-Are the figures (Tables, Images) of sufficient quality for clarity?

Reviewer #1: See general and specific comments

Reviewer #2: (No Response)

**Conclusions**

-Are the conclusions supported by the data presented?

-Are the limitations of analysis clearly described?

-Do the authors discuss how these data can be helpful to advance our understanding of the topic under study?

-Is public health relevance addressed?

Reviewer #1: See general and specific comments

Reviewer #2: (No Response)

**Editorial and Data Presentation Modifications?**

Reviewer #1: Include S4 Appendix in the manuscript

Reviewer #2: (No Response)

**Summary and General Comments**

Reviewer #1: General comments

In order to add new knowledge through the review, the following points should be taken in consideration in a revised manuscript:

- For 1538 cases, there was only 106 isolation of Brucella spp, i.e. less than 7%. Therefore, all the discussion related to the importance of Brucella species per worker category should be introduced by a word of caution

- Specify per worker category the number of Brucella isolation.

- A specific section devoted to human brucellosis due to RB51 should be included, although the vast majority of such cases are linked to the consumption of raw milk.

- Why is B. canis not discussed in this review although it appears in figure 2?

- Laboratory workers: specify the importance of this group in countries where brucellosis has been eradicated. In addition, it would be worth to mention that mis diagnosis (as Ochrobactrum antropi) occurs regularly

Consider including S4 Appendix in the manuscript.

Although the systematic review is proficiently executed, the information is to a large extent not new. Therefore, the authors should discuss in dept which new information has been generated through their work. In this perspective, the meta-analysis is perhaps the most interesting part of this research. 

Specific comments

Title: not Ok, I suggest: Occupational exposure to Brucella spp.: a systematic review and meta-analysis

L27: live attenuated anti-brucellosis vaccines

L41: different wildlife species

L45: live attenuated anti-brucellosis vaccines

L69-72: introduce the DALY concept

L76: live attenuated anti-brucellosis vaccines

L167-169: specify to which animal species the aborted fetus belongs

L189: live attenuated anti-brucellosis vaccines

L222: reports on contamination by b. suis biovar 2 in France should be mentioned in this section

L258-264: the explanation is imported cases of brucellosis in non-endemic countries, not scientific interest

L271: B. abortus is the most important species isolated in ref 86. However, other references, particularly from China show that B. melitensis is the most important species, whereas in Kazakhstan, B. melitensis is exclusively isolated from human patients

L272-274: the symptoms are for all Brucella species. It is thus incorrect to ascribe them to B. abortus

L343: all systematic review should follow the PRISMA statement

274: Brucella melitensis in not more pathogenic than B. abortus in humans. It is just the most prevalent species!

L281: reference

L304: laboratory worker: see general comment

L325: hunters: for which countries is this group important?

L293: it should be highighted her and in the abstract that veterinarians are th groups for which exposure to vaccine strains is the most important

L326-328: this is a wrong statement. B. suis infection is mainly due to contact and dressing of carcasses, not consumption of meat

L340: wording: expressive? 

L350: L354: The Ine Health concept comes out of the blue and should either be deleted or better explained

Figure 2: this figure may give a wrong impression because the status towards brucellosis has changed (several countries became "officially free" from brucellosis) during the study period. For example, for Germany, Canada, USA abattoir workers are mentioned although this is not true anymore. In addition, an explanation must be given on why only veterinarians in India and only abattoir workers in Nigeria are reported while many rural worker are reported in other African countries. Answers to this type of questions would possibly generate new information.

Reviewer #2: See attached document

PLOS authors have the option to publish the peer review history of their article (what does this mean?). If published, this will include your full peer review and any attached files.

Reviewer #1: Yes: Jacques Godfroid

Reviewer #2: No

---

## [Editor Report · Decision Letter 1]

21 Feb 2020

Dear Dr. Dorneles,

We are pleased to inform you that your manuscript 'Occupational exposure to Brucella spp.: a systematic review and meta-analysis' has been provisionally accepted for publication in PLOS Neglected Tropical Diseases.

Before your manuscript can be formally accepted you will need to complete some formatting changes, which you will receive in a follow up email. A member of our team will be in touch within two working days with a set of requests.

Best regards,

Tao Lin, DVM, MSc

Associate Editor

Elsio Wunder Jr

Deputy Editor

---

## [Editor Report · Acceptance letter]

30 Apr 2020

Dear Dr. Dorneles,

We are delighted to inform you that your manuscript, " Occupational exposure to *Brucella* spp.: a systematic review and meta-analysis ," has been formally accepted for publication in PLOS Neglected Tropical Diseases.

Best regards,

Serap Aksoy

Editor-in-Chief

Shaden Kamhawi

Editor-in-Chief
